# Vanadium: A Review of Different Extraction Methods to Evaluate Bioavailability and Speciation

**Jie Yang** [1,2]**, Yunlong Wang** [1,2]**, Xiaohui Gao** [1,2]**, Rui Zuo** [1,2,*]**, Liuting Song** [1,2,*]**, Chenhui Jin** [1,2]**, Jinsheng Wang** [1,2] **and Yanguo Teng** [1,2]

1    College of Water Sciences, Beijing Normal University, Beijing 100875, China; yangjie@bnu.edu.cn (J.Y.); wangyl0115@163.com (Y.W.); gaoxh4823@163.com (X.G.); jchh322@163.com (C.J.); wangjs@bnu.edu.cn (J.W.); ygteng@bnu.edu.cn (Y.T.)
2    Engineering Research Center of Groundwater Pollution Control and Remediation, Ministry of Education, Beijing 100875, China
*    Correspondence: zr@bnu.edu.cn (R.Z.); ltsong@bnu.edu.cn (L.S.)

**Abstract:** The excessive input of heavy metals such as vanadium (V) into the environment has been one of the consequences of global industrial development. Excessive exposure to V can pose a potential threat to ecological safety and human health. Due to the heterogeneous composition and reactivity of the various elements in soils and sediments, quantitative analysis of the chemical speciation of V in different environmental samples is very complicated. The analysis of V chemical speciation can further reveal the bioavailability of V and accurately quantify its ecotoxicity. This is essential for assessing for exposure and for controlling ecological risks of V. Although the current investigation technologies for the chemical speciation of V have grown rapidly, the lack of comprehensive comparisons and systematic analyses of these types of technologies impedes a more comprehensive understanding of ecosystem safety and human health risks. In this review, we studied the chemical and physical extraction methods for V from multiple perspectives, such as technological, principle-based, and efficiency-based, and their application to the evaluation of V bioavailability. By sorting out the advantages and disadvantages of the current technologies, the future demand for the in situ detection of trace heavy metals such as V can be met and the accuracy of heavy metal bioavailability prediction can be improved, which will be conducive to development in the fields of environmental protection policy and risk management.

**Keywords:** extraction methods; chemical speciation; bioavailability; vanadium



## 1. Introduction

Vanadium (V) has many valuable physical, chemical, and mechanical properties; therefore, it is widely used in modern industrial technologies and is an important strategic material [1]. V is used as an additive and alloying element (at 80–85%) in the ferrous metallurgy industry to prepare a unique kind of steel. In the chemical industry, V compounds have been widely used as catalysts and cracking agents in the contact method sulfuric acid manufacturing industry and petroleum refining and organic synthesis industries [2,3]. A large number of studies have shown that excessive V is toxic and carcinogenic and must be treated at the same level as Pb, As, and Hg [4–7]. As early as the end of the 1980s, the United Nations Environment Programme (UNEP) recommended that V be included in the list of environmentally hazardous elements as a substance of priority. Moreover, they jointly proposed with the World Health Organization (WHO) and the International Agency for Chemical Substances Safety (IPCS): "On the basis of monitoring, research on the environmental behavior and biological toxicological characteristics of V should be strengthened" [8,9].

Early V was mainly stored in the mantle and was later released into the surface environment through the combustion of fossil fuels, mining, and the use of phosphate

fertilizers [5]. V is chemically active and can form species with variable oxygen affinities. On entering the soil, V and its compounds undergo chemical migration via various reactions, such as dissolution, precipitation, aggregation, complexation and adsorption [10]. Both the cationic ($V^{2+}$, $V^{3+}$, $VO^{2+}$, $VO_2^+$, $V(OH)^{2+}$, and $V(OH)_4^+$) and anionic states ($VO_3^-$, $VO_4^{3-}$, $HVO_4^{2-}$, $H_2VO_4^-$, $VO_2(OH)^-$, $VO_3(OH)^{2-}$, and $V_4O_{12}^-$) of V are toxic, and their toxicity also depends on the physical and chemical properties of the compounds they form [11–13]. In recent years, monitoring the total concentration of V has remained useful in many areas, but speciation research is of vital importance because the mobility, bioavailability, bioaccumulation, and toxicity of V depend on its chemical species [14–16]. The determination of the speciation of V in different environmental media, such as in river sediments, mining area soils, and tailings, has been widely reported [17–21]. Similar to other chemical poisons, V toxicity in different environmental media generally increases with increasing atomic valence state and solubility [22,23].

The main principle of the current extraction methods from soils is using different chemical reagents or solvents to separate and test V based on differences in the physical properties (such as particle size, solubility, etc.) or chemical properties (such as binding states, reactivity, etc.) of the different species of V [2,24–33]. Due to the different extraction mechanisms, the extraction outcomes from soils using various extractants are quite different [34,35]. Correlations between the amount of V extracted by different methods from soils and the V absorbed by plants can reflect the bioavailability of V [16,24]. Good correlations could predict the amount of V absorbed by plants, thereby indicating its biological impact [36,37]. The amount of V species absorbed by the plant should be correlated with the amount of extracted V species by statistical analysis to establish the bioavailability of V species in the soil. At present, a comprehensive understanding of the selectivity of the extractants, their effect on the chemical speciation of V in different environmental media, and the evaluation of the bioavailability of V to plants is still lacking. Therefore, this review aims to summarize and discuss globally published studies from the past ten years on the extraction of specific chemical species of V. The objectives of this review are (1) to review and compare the most relevant and recently published extraction methods and extraction efficiencies of V from different environmental samples worldwide and (2) to summarize the use of extraction methods related to the various chemical species of V to assess the bioavailability of different V species on plants. This review will provide a more systematic and specific discussion of the accurate evaluation of the bioavailability and risks associated with V in the environment, which will further lay the foundation for revealing its geochemical behavior.

## 2. Sources and Human External Exposure Pathways

### 2.1. The Distribution of V Resources

V is widely distributed in nature, with an average content of 90 mg kg$^{-1}$ and 0.0015% mass ratio in the Earth's crust, a concentration higher than those of Cu, Ni, Zn, Ti, Co, Pb, and other metals [5,15]. The valence or oxidation state of V determines the properties of the compounds it forms. Common V oxides include VO, $V_2O_3$, $VO_2$, and $V_2O_5$. Their oxidation states range from low (II) to high (V), their redox abilities also change from strongly reducing to strongly oxidizing, and their aqueous solutions gradually change from strongly alkaline to weakly acidic [13,38].

In nature, V mainly forms symbiotic or composite ores with other minerals. More than 70 kinds of V-bearing minerals have been discovered at present and few individual V deposits have high content or rich accumulation [39]. Most V deposits are associated with V-Ti magnetite, potash V uranium, and petroleum-associated minerals. Ninety-eight percent of the proven V resource reserves are contained in V-Ti magnetite and their $V_2O_5$ content is 1.8% [5]. The world's total metal V resources are estimated to be approximately 41.3 million tons. In the order of reserves, at present, the main V-supplying countries of the world are Russia, South Africa, China, the United States, and Australia [40]. The V concentration values of some countries in the world are shown in the Table 1. The V

concentration in some areas of Turkey, Spain, and other countries is close to the world average level by 90 mg·kg$^{-1}$.

**Table 1.** V concentrations in some countries of the world.

| Region | V Concentration in Soils (mg/kg) | Data from Reference |
|---|---|---|
| Poland | 18.39 | [41] |
| Palermo, Italy | 58 | [42] |
| Cheppel Island, Hungary | 15.2–42.0 | [43] |
| Catalonia, Spain | 15.2–144.9 | [44] |
| Arcala de Enares, Spain | 6.01 | [45] |
| Turku, Finland | 47.5 | [46] |
| Lithuania | 38 | [47] |
| Russia | 79–91 | [48] |
| Ankara, Turkey | 74 | [49] |

China is rich in V resources, and its average V soil content is 114 mg/kg, which is 27% higher than the worlds' average [36,50]. China's V resources are mainly distributed in seven provinces (>90%) (Table 2), of which the Sichuan Province ranks first in the country, accounting for 49% of the total reserves [22,51]. Among them, V-Ti magnetite is mainly distributed in the Panzhihua-Xichang region, Sichuan Province, and black shale-type V deposits are mainly distributed in the Hunan, Hubei, Anhui, and Jiangxi provinces [27,52]. There are two main forms of V ore resources in China: (1) those produced in magmatic rock-type V-Ti magnetite deposits as associated minerals and (2) independent deposits, mainly Cambrian black shale-type V ore. In addition, China also has abundant stone coal and V resources. Although stone coal V ore is a low-grade V-bearing resource, its V content is equivalent to the world's total reserves of non-stone coal V ore resources.

**Table 2.** V concentrations in contaminated soils of China.

| Region | V Concentration in Soils (mg/kg) | Data from Reference |
|---|---|---|
| Chongqing | 39–4994.6 | [53] |
| Sichuan Province | 19.1–548.7 | [53] |
| Chengdu city, Sichuan Province | 66.69–73.25 | [54] |
| | 149–4794 | [51] |
| Panzhihua city, Sichuan Province | 167 | [52] |
| | 71.1–938.4 | [36] |
| | 105.57–189.12 | [50] |
| | 1120.3–1139.9 | [55] |
| Hunan Province | 4.5–1390.8 | [53] |
| | 62.81–152.77 | [55] |
| Chenxi county, Huaihua city, Hunan Province | 168–1538 | [27] |
| | 1500–2600 | [56] |
| Lianyuan city, Hunan Province | 38.97–618.90 | [57] |
| Loudi city, Hunan Province | 97–282 | [58] |
| Hubei Province | 17.6–836.2 | [53] |
| | 500 | [59] |
| | 1306 | [60] |
| | 931 | [4] |
| Shiyan city, Hubei Province | 1998.7–2031.5 | [61] |
| | 128–821 | [6] |
| Shaanxi Province | 26.6–1854 | [53] |
| | 85.98 | [62] |
| | 21.14–286.42 | [63] |
| Langao county, Ankang city, Shaanxi Province | 264–596 | [64] |
| Xi'an city, Shaanxi Province | 53.9–89.7 | [65] |

**Table 2.** *Cont.*

| Region | V Concentration in Soils (mg/kg) | Data from Reference |
|---|---|---|
| | 85.2 | [66] |
| Anhui Province | 23.3–1746.6 | [53] |
| Huainan city, Anhui Province | 2.24–71.86 | [67] |
| | 41.15–81.13 | |
| Yunnan Province | 6.7–1546 | [53] |
| | 168 | [53] |
| Kunming city, Yunnan Province | 281.56 | [68] |
| Guizhou Province | 16–1685 | [53] |
| Bijie city, Guizhou Province | 206 | [69] |
| Zunyi city, Guizhou Province | 170–1369 | [70] |

*2.2. V Contamination Sources in Soils*

The smelting of V and its alloys are the main pollution sources associated with V in the environment. V is discharged into the environment during a series of processes, such as mining, crushing, sintering, and steelmaking from V-containing minerals such as V-Ti magnetite [18,19,36,53]. At present, the areas with severe V pollution are mainly concentrated in industrial areas, thermal power plants, V-Ti magnetite mines, smelters, etc., that use heavy oil and coal as fuel. Generally, V pollution is relatively serious near thermal power plants, which burn 20 to 30 tons of heavy oil per hour and discharge 20 kg of V pentoxide [71]. In winter and spring, 50% of the V pollution in the Russian Arctic is caused by V deposition from the atmosphere [72]. Part of the open-pit V mines also discharge into rivers and farmlands with surface runoff. At the same time, due to the recharge of V-containing wastewater and the application of V-containing pesticides, the V content in farmland soil far exceeds the background value of heavy metals in this area, thereby threatening human health [27,35,73,74]. The impact of V mining on the ecological environment is mainly reflected in topography, land occupation, soil erosion, and so on. Its biological effects are mainly manifested as the destruction of animal populations and vegetation. Societal impacts are mainly manifested as changes in land use patterns and landscape patterns. Panzhihua city and the Greater Western Hunan Region in China are rich in V ore resources, and a large number of V smelting projects have caused V contamination, which severely endangers the ecology of these areas [6,27,28,36,51,75].

Agricultural production, especially the excessive use of heavy metal-contaminated fertilizers, organic manures, urban waste and pesticides, and sewage irrigation in modern agricultural production processes, can cause heavy metal contamination in agricultural soils [36,76,77]. The long-term, improper application of fertilizers can not only lead to soil acidification and nutrient ratio imbalances but also promote the release of toxic and harmful pollutants [78]. Affected by the deposition processes of phosphate ores, phosphate fertilizers often contain a large quantity of heavy metals, and the heavy metal content depends on the choice of phosphate ore sedimentary facies and manufacturing processes used to prepare phosphate fertilizers [79,80]. Studies have shown that V is widely present in fertilizer products, its content in general pesticides can be as high as 45%, and the content of V in farm manure can reach 3–8 mg/kg [81,82]. Among these sources, the content of V in phosphate fertilizers is relatively high, mostly existing in the form of the soluble and most toxic pentavalent V salts, such as $NH_4VO_3$, which has potential environmental risks [79,80]. In addition, the livestock and poultry breeding industries in agricultural production are also important sources that cannot be ignored [78]. The excess V is applied to farmland soils in the form of organic fertilizers [80,83].

Studies have shown that the amount of V that enters the soil globally through fertilizers can be as high as 1500 tons per year, which poses a threat to farmland ecosystems [78]. In addition to the large proportion of lead, cadmium, arsenic, and other elements that exceed the standards in the soils of many grain-producing areas, the content of V has also been significantly higher than its background levels [22,28,84]. Once heavy metal

pollution occurs in farmlands, it is very difficult to control due to the large areas involved. The V accumulated in the surface soils continuously enters the underground environment through irrigation and rainfall. Due to its refractory nature and biological toxicity, it poses a severe threat to human health and ecological safety.

### 2.3. V Speciation in Soil

V has a complex species in soil and can be combined with other metal ions and soil organic matter to form a variety of chemical species. V has various oxidation states (+2, +3, +4, +5), but in the natural environment it still exists mainly in the species of V(IV) and V(V) [5,85]. The morphology of V largely depends on the redox conditions of the environment, and different redox pairs ($NO_3^-$/$NH_4^+$, $Fe^{3+}$/$Fe^{2+}$, $MnO_2$/$Mn^{2+}$ and $SO_4^{2-}$/$H_2S$) play important roles in the transformation of V species. Under oxidative and moderately reducing conditions, V(IV) and V(V) dominate [28]. V(IV) is stable under acidic conditions (pH < 5), but V(IV) is gradually oxidized to V(V) with increasing pH [10]; conversely, V(V) can also be reduced to relatively unstable V(IV) by humic substances, hydrogen sulfide, and other soil organic components (SOM) [26,85,86]. V(III) exists only in strictly anaerobic environments, such as some primary minerals, saturated soil, or peat, and is easily oxidized to V(IV) and V(V) [10]. Under normal circumstances, V(V) mostly exists in the species of anions ($H_2VO_4^-$ and $HVO_4^{2-}$) with strong mobility, while oxides/hydroxides of iron, aluminum, and manganese in the soil can combine with it to reduce V(V) fluidity; while V(IV) usually appears in the species of $VO^{2+}$, which can bind to organic ligands and is relatively stable under relatively reduced and low pH environments [87,88]. Soil pH also significantly affects the chemical species of heavy metals in soil solutions [29], and studies have reported that increasing soil pH may promote the migration and release of V in soil [89,90]. Changes in soil pH may also control the morphological changes, mobility, and bioavailability of V by affecting the solubility of SOM. Oxygen-containing functional groups undergo deprotonation with increasing pH, which in turn increases the solubility of SOM [10]. SOM can promote reducing conditions and help reduce V(V) to V(IV) [91]; on the other hand, SOM can also stimulate the proliferation of microorganisms, thereby promoting bioreduction [10,28]. Panichev (2006) analyzed V(IV) and V(V) in soil and plants of V-contaminated sites, and confirmed that V in soil and plants mainly exists in the species of +5 valence, which is potentially harmful [92]. $V_2O_5$ is more soluble and more toxic than $V_2O_3$ and $VO_2$, so the valence of V is more meaningful than the total amount of V. In soil, V's mobility, bioavailability and ecotoxicity in soil-plant systems are closely related to its chemical species. Residual V does not participate in a series of biochemical processes in the soil, but it can cause potential risks to the soil through reactions such as hydrolysis, oxidation, and reduction. Changes in soil redox potential, pH, organic matter content, metal oxide (hydroxide) content, and microbial activity all affect the morphological changes of V [28,38,93].

## 3. The Extraction Methods for V

### 3.1. Single Extraction Methods

Single extraction methods involve the mixing of one or several mixed reagents with soil in a specific ratio of soil to extractant liquid. Through one-step leaching, the content of a specific species of V in the solution is determined, and the extractable concentration of V has good correlations with its content in plants [25]. Because single extractions are relatively fast, inexpensive, and easy to conduct, various methods are widely used to assess V mobility in soils and sediments and to evaluate the short-term or medium-term hazards of heavy metals [11,26,36,94–96]. The extraction agents mainly use ion exchange, dissolution (acid or alkali), or chelation to extract various species of V from soil [11,26,96]. Commonly used leaching agents are as follows: (1) weak (dilute) acids (0.1 M HCl, 0.1 M $HNO_3$, 0.5 M HOAc, etc.); (2) chelating agents (DTPA, EDTA, etc.); and (3) inorganic salt solutions, including valence cation salt solutions ($NH_4OAc$, $NH_4NO_3$, $NaNO_3$, etc.) and divalent cation salt solutions ($CaCl_2$, $BaCl_2$, etc.) [16,26–28,36,76,91,97].

Dilute acid solutions are mainly used for acidic soil, and dilute HCl is the most commonly used reagent. An intermediate concentration of HCl (0.5 M) has buffering capacity, mainly dissolving metals in carbonates, and has a limited impact on the metals in residual clays and sulfides [98]. The leaching capacity of hydrochloric acid for V is between that of the chelating agents and salt solutions [36,99]. The V extracted by HCl has a positive correlation with the V in plants [36]. Furthermore, the amount of HCl leaching V has a significant positive correlation with the initial concentration of HCl (5–30%) and the reaction temperature (80–110 °C) [27]. Nevertheless, the impacts of soil properties, including pH, CEC, and TOC, on V extractions with HCl are not obvious [36]. A complexing agent can form very stable, water-soluble, and well-defined complexes with metal ions and can simulate the activation of heavy metals by plant root exudates [100,101]. Among them, ethylenediamine tetraacetic acid (EDTA) (0.05–1 M) and diethylenetriaminepentaacetic acid (DTPA) (0.005 M) are commonly used chelating agents [19,20,36,102–104]. Taking EDTA as an example, it is a nonselective reagent and can exhibit a strong capacity for complexing metals [19]. It can dissolve carbonates and form organometal complexes, which compete with the organic matter in soils. Additionally, complexes of EDTA with heavy metals such as Pb, Zn, Cd, Cu, and Ni are more stable than complexes between EDTA and V, which may result in a lower EDTA-extractable concentration of V when the aforementioned heavy metal concentrations are high [74]. Moreover, the complexing agent often forms a mixed solution with inorganic salts and dilute acids to improve the extraction efficiency.

Inorganic salt solutions mainly extract heavy metals in water-soluble and exchangeable states through ion exchange [105]. Among them, $CaCl_2$ and $NaNO_3$ are also often adopted as extractants for chemical speciation prediction [101]. Research has shown that $NaNO_3$ (0.1 M) and $CaCl_2$ (0.01 M) extraction methods are only suitable for exchangeable metals [16,19,50,100,101,106]. The ionic strength of $NaNO_3$ is similar to that of the soil solution, so it cannot affect the equilibrium between soil solids and soil solutions [107]. In a previous study, EDTA, HCl, and $NaNO_3$ were used as comparative extractants to evaluate V levels in the rhizospheric soil of alfalfa [16,50]. Out of the three extractants, HCl extracted the highest V concentrations (4.75–307.84 mg/kg) from soils where EDTA extracted 3.15–393.61 mg/kg V and $NaNO_3$ extracted 0.004–23.94 mg/kg V from soils. Normally, $NaNO_3$ exerts weak competition for the adsorption sites of oxide surfaces and organic matter [108]. Therefore, the weak leaching capacity of $NaNO_3$ was the main reason for its lower extraction of V compared to the dilute acid and complexing agent.

In addition, $CaCl_2$ is often used for the extraction of V because calcium is an important cation in soils and reflects the differences in solubility or binding strength between different soils [19,101,109,110]. The mobile V in the alluvial soils of Belgium in Europe was estimated by a single extraction with $CaCl_2$ (0.01 M) [11]. According to calculations, the main V species encountered in the $CaCl_2$ extracts is $HVO_4^{2-}$, which means that V occurs as an anion that will have the tendency to be desorbed when the solution pH rises. V extraction was very low even in the most contaminated soil samples of this experiment, which indicated its low mobility. However, it is necessary to study the impact of changing environmental conditions such as fluctuating redox conditions and soil acidification [110].

Due to the different extraction mechanisms, the extraction outcomes of different extractants are quite different. Moreover, different leaching agents are suitable for soils in different environments [19]. Because soil pH has a substantial impact on the mobility of heavy metals, the use of different extractants for soils at different pH values can improve extraction efficiency. According to previous studies, the extractants that can better predict the migration characteristics of heavy metals in different kinds of soils are listed in Table 3. Among them, the heavy metals extractable by the DPTA, $NaNO_3$, NaAc, and $CaCl_2$ leaching methods correlated well with the content of V in plant roots for neutral and near-alkaline soils [25,109]. Positive correlations can be observed in acidic soil for the EDTA, HCl, $CaCl_2$, $NH_4Ac$, $Ca(NO_3)_2$, and $NaH_2PO_4$ leaching methods [37,100,101,111–113]

**Table 3.** Comparison of the V extracted by single-step extraction methods.

| Extractant Category | Extraction Solutions | Extraction Yield (Extracted V/Total V) | Samples | Data from Reference |
|---|---|---|---|---|
| Weak (dilute) acids | HCl | 2.73 ± 2.21% | 0.01 M HCl, soil samples collected from Zhujiabaobao mine located in the eastern part of Panzhihua mine area ($n = 7$, $V_{total} = 67.43 \pm 14.92$ mg/kg) | [25,28] |
| | | 6.21–69.26% | 0.5 M HCl, soil used in pot experiment was collected from moist soil (0–20 cm) of forest land in an urban park in Panzhihua, southwest China ($n = 75$, $V_{total} = 7.73–494.45$ mg/kg) | |
| | HNO$_3$ | 2.68 ± 1.65% | 0.43 M HNO$_3$, soil samples collected from Zhujiabaobao mine located in the eastern part of Panzhihua mine area ($n = 7$, $V_{total} = 67.43 \pm 14.92$ mg/kg) | [28] |
| | HOAc | 0.01–1.33% | 0.11 M HOAc, topsoil (0–10 cm) samples were collected from the Panzhihua, urban park ($n = 23$, $V_{total} = 105.57–189.72$ mg/kg) | [36] |
| | Citric acid (C$_6$H$_8$O$_7$) | 2.39 ± 2.03% | 0.1 M C$_6$H$_8$O$_7$, soil samples collected from Zhujiabaobao mine located in the eastern part of Panzhihua mine area ($n = 7$, $V_{total} = 67.43 \pm 14.92$ mg/kg) | [28] |
| Chelating agents | EDTA | 0.2–35% | 0.025 M Na$_2$-EDTA, soil samples from different sites of the German long-term soil monitoring program ($n = 30$, $V_{total} = 1.7–143.0$ mg/kg) | [25,36,74] |
| | | 4.33–61.98% | 0.05 M EDTA, soil used in pot experiment was collected from moist soil (0–20 cm) of forest land in an urban park in Panzhihua, southwest China ($n = 75$, $V_{total} = 7.73–494.45$ mg/kg) | |
| | | 0.27–4.09% | 0.05 M EDTA, topsoil (0–10 cm) samples were collected from the Panzhihua, urban park ($n = 23$, $V_{total} = 105.57–189.72$ mg/kg) | |
| | DTPA | 0.6–7.7% | 1 M NH$_4$HCO$_3$ + 0.005 M DTPA, cultivated soils of Egypt and Greece ($n = 21$, $V_{total} = 13–206$ mg/kg) | [91,114] |
| | | 0.37–5.12% | 1 M NH$_4$HCO$_3$ + 0.005 M DTPA, different types of soil sampled from three different study areas in Germany ($n = 6$, $V_{total} = 29.7–109.2$ mg/kg) | |

**Table 3.** *Cont.*

| Extractant Category | Extraction Solutions | Extraction Yield (Extracted V/Total V) | Samples | Data from Reference |
|---|---|---|---|---|
| Inorganic salt solution | NaNO$_3$ | 0.005–4.84% | 0.1 M NaNO$_3$, soil used in pot experiment was collected from moist soil (0–20 cm) of forest land in an urban park in Panzhihua, southwest China ($n = 75$, V$_{total}$ = 7.73–494.45 mg/kg) | [25] |
| | CaCl$_2$ | <4% | 0.01 M CaCl$_2$, seventeen rural soil profiles for this study were selected to covera representative range of different parent materials in Taiwan ($n = 94$, V$_{total}$ = 35.4–475 mg/kg) | [76] |
| | NaHCO$_3$ | <4% | 0.5 M NaHCO$_3$, seventeen rural soil profiles for this study were selected to covera representative range of different parent materials in Taiwan ($n = 94$, V$_{total}$ = 35.4–475 mg/kg) | [76] |

### 3.2. Sequential Extraction Methods

Sequential extraction (SE) methods are well-established approaches for analyzing the chemical speciation of V. SE is a process of classified extraction by determining one or a group of substances from the sample according to their physical properties (such as particle size, solubility, etc.) or chemical properties (such as binding states, reactivity, etc.). In recent years, many SE methods have been used to extract V, as summarized in Table 4. As early as 1979, Tessier et al. (1979), based on geochemical characteristics and simulated common environmental conditions, invented a five-step sequential extraction to divide the chemical speciation of elements in the soil into acid-soluble states (including water-soluble states), carbonates, Fe and Mn oxides, organic matter, and residual states [115]. Based on this method, many scholars have analyzed the chemical speciation of V in contaminated soils or sediments in China, the USA, Italy, Spain, Turkey, Poland and U.K. [99,116–121]. Most morphological analyses results indicate that V is mainly present in the soil as the residual fraction (70–85%), and present in lower quantities in Fe and Mn oxides (6.96–18.0%) or organic matter (2.48–13.2%) [20,116,118,119]. In contrast with these studies, significantly larger amounts of V were found in the Fe and Mn oxide fractions (22.4 and 78% of the total, respectively) in some other studies [117,121]. The residual fraction only contained 8 to 51.8%, while the organic matter fraction was of minor importance, accounting for 3–22.4% of the total V in the soil. The large specific surface area of oxide and hydroxide clay minerals in soils has more adsorption sites, and the different contents of iron, manganese, aluminum oxides, and hydroxides in different soils are important factors leading to the inconsistency of various species of soil V [121].

**Table 4.** Comparison of the V extracted from different environment samples by sequential extraction methods.

| Samples | Location | Sequential Extraction Methods (SE) | Target Fraction | Extraction Solutions | Average Percentage of Each Fraction (%) | Data from Reference |
|---|---|---|---|---|---|---|
| Soil | Agriculture region, Panzhihua city, Sichuan province, China. $V_{total}$: 71.7–938 mg/kg, $n = 55$ | Wenzel scheme | Non-specifically sorbed | 0.05 M $(NH_4)_2SO_4$, 20 °C | 0.51 | [99] |
| | | | Specifically sorbed | 0.05 M $NH_4H_2PO_4$, 20 °C | 0.30 | |
| | | | Amorphous hydrous Fe and Al oxide | 0.2 M $NH_4^+$-oxalate buffer, pH = 3.25, 20 °C | 5.52 | |
| | | | Crystalline hydrous Fe and Al oxides | 0.2 M $NH_4^+$-oxalate buffer + Ac, pH = 3.25, 96 °C | 9.83 | |
| | | | Residual | $HNO_3$ + $H_2O_2$ (1:50) | 83.80 | |
| | | Five-steps SE of Tessier | Exchangeable | 1 M $MgCl_2$, pH = 7, 20 °C | 0 | [122] |
| | | | Carbonates | 1 M HOAc + NaOAc, pH = 5, 20 °C | 0.14 | |
| | | | Fe and Mn oxides | 20 mL 0.04 M $NH_2OH·HCl$ in 25% ($v/v$) HOAc, 96 °C | 6.96 | |
| | | | Organic matter | 0.02 M $HNO_3$ + 30% $H_2O_2$ (pH = 2), 85 °C 3.2 mol/L $NH_4OAc$ in 20% $HNO_3$ | 13.20 | |
| | | | Residual | HF + $HClO_4$ (5:1) | 79.70 | |
| | Agricultural region, In South Finland from bare arable land, $V_{total}$ = 601 mg/kg, $n = 1$ | Modified on the basis of procedures used in the sequential frac-tionation of selenium (Se) and phos-phorus (P) | Easily soluble V | 0.25 M KCl | <11 | [29] |
| | | | V bound by ligand exchange | 0.1 M $KH_2PO_4$ + $K_2HPO_4$ | 8~35 | |
| | | | Organic V | 0.1 M NaOH | 30~68 | |
| | | | Strong bound V | 0.25 M $H_2SO_4$ | <10 | |
| | Agricultural region, Eschikon, Switzerland. $V_{total}$ = 61.2 mg/kg, $n = 1$ | Sequential extraction of V in soils was performed based on Wenzel et al. (2001) before soybean planting | Non-specifically sorbed | 0.05 M $(NH_4)_2SO_4$, 20 °C | 15.70 | [20] |
| | | | specifically-sorbed | 0.05 M $NH_4H_2PO_4$, 20 °C | 24.60 | |
| | | | Amorphous and poorly crystalline hydrous oxides of Fe and Al | 0.2 M $NH_4^+$-oxalate buffer, pH = 3.25, 20 °C | 23.80 | |
| | | | Well-crystallized hydrous oxides of Fe and Al | 0.2 M $NH_4^+$-oxalate buffer + AC, pH = 3.25, 96 °C | 3.80 | |
| | Mining region, Panzhihua city, Sichuan province, China. $V_{total}$ = 69.8–279.35 mg/kg, $n = 7$ | BCR SE | Acid soluble | 0.11 M $CH_3COOH$, 25 °C | 0.19–0.82 | [28] |
| | | | Reducible | 0.5 M $NH_2OH·HCl$ + 0.05 M $HNO_3$, 25 °C | 0.27–5.78 | |
| | | | Oxidizable | 30% $H_2O_2$, pH = 2, 85 °C; 1 M $NH_4OAc$, pH = 2, 25°C | 4.37–10.50 | |
| | | | Residual | $HNO_3$ + $H_2O_2$ (1:50) | 83.3–93.10 | |
| | Mining region, Chenxi county, Hunan province, China. $V_{total}$ = 168–1538 mg/kg, $n = 7$ | Modified BCR SE | Acid extractable | 0.11 M HOAc | 0.32–1.88 | [27] |
| | | | Reducible | 0.5 M $NH_2OH·HCl$, pH = 1.5 | 5.63–34.40 | |
| | | | Oxidizable | 8.8 mol/L $H_2O_2$; 1 M $NH_4OAc$, pH = 2 | 0.81–22.90 | |
| | | | Residual | $HNO_3$ + $H_2O_2$ (1:50) | 57.70–58.80 | |
| | Industrial region, Milazzo area, Sicily, $V_{total}$ = 0.072–0.24 g /kg, $n = 23$ | Modified Tessier's SE | Exchangeable V | 1 M NaOAc | 2.19 | [73] |
| | | | V bound to carbonates | $CH_3COONa/CH_3COOH$, pH =5 | 1.19 | |
| | | | V bound to Fe and Mn oxides | 0.04 M $NH_3(OH)Cl$ + 25% $CH_3COOH$ ($v/v$), 96 °C | 0.88 | |
| | | | V bound to organic matter and or sulfide | $HNO_3$ + 30% $H_2O_2$ | 92.31 | |
| Sediment | Nile Delta coast, $V_{total}$ = 73.62–154.82 mg/kg, $n = 11$ | Modified BCR SE | Exchangeable | 1 M $NH_4CH_3COO$, pH = 7 | 7.20 | [123] |
| | | | Acid-reducible | 0.25 M $NH_2OH·HCl$, pH = 2 | 1.60 | |
| | | | Oxidizable-organic | 30% $H_2O_2$, 1 M $NH_4CH_3COO$, pH = 2 | 27.40 | |
| | | | Resistant | 65% $NHO_3$ + 70% $HCLO_4$ + HF | 63.80 | |
| | In northern part of Belgium. $V_{total}$ = 40–430 mg/kg, $n = 14$ | Modified BCR SE | Acid soluble | 0.11 M $CH_3COOH$ | 0.14 | [17] |
| | | | Reducible | 0.1 M $NH_2OH·HCl$, pH = 1.5 | 21.16 | |
| | | | Oxidizable | 8.8 M $H_2O_2$, 1.0 M $NH_4OOCH_3$ | 9.72 | |
| | | | Residual | HCl + $HNO_3$ + HF (2:1:1) | 52.93 | |

**Table 4.** *Cont.*

| Samples | Location | Sequential Extraction Methods (SE) | Target Fraction | Extraction Solutions | Average Percentage of Each Fraction (%) | Data from Reference |
|---------|----------|-----------------------------------|-----------------|----------------------|-----------------------------------------|---------------------|
| Ore (coal) | Industrial region, Anatolia, Turkey. $V_{total}$ = 701, $n$ = 1. | Seven-step sequential extraction procedure of the coal bottom ash | Water soluble | Deionized water | 1.51 | [124] |
| | | | Exchangeable fraction | 1 M $MgCl_2 \cdot 6H_2O$, pH = 7 ± 0.1 | 1.28 | |
| | | | Carbonate fraction | 1 M NaAc, pH = 5 ± 0.1, 90 °C | 12.13 | |
| | | | Reducible fraction | 0.1 M $NH_2OH \cdot HCl$ + 25% ($v/v$) $CH_3COOH$, 90 °C | 25.11 | |
| | | | Oxidizable fraction | $H_2O_2$, pH = 2 ± 0.1, 100 °C | 7.13 | |
| | | | Sulfide fraction | Aqua regia, 120 °C | 25.11 | |
| | | | Residual | HF + HCl + $HNO_3$ (5:1:5) | 25.76 | |
| Ore (asphaltite) | Minging region, in SE Anatolia of Turkey. $V_{total}$ = 546.15 mg/kg, $n$ = 1. | Seven-step sequential extraction procedure of asphaltite combustion waste | Water soluble | Deionized water | 1.66 | [125] |
| | | | Exchangeable fraction | 1 M $MgCl_2 \cdot 6H_2O$, pH = 7 ± 0.1 | 2.61 | |
| | | | Carbonate fraction | 1 M NaAc, pH = 5 ± 0.1, 90 °C | 6.00 | |
| | | | Reducible fraction | 0.1 M $NH_2OH \cdot HCl$ + 25% ($v/v$) $CH_3COOH$, 90 °C | 11.75 | |
| | | | Oxidizable fraction | $H_2O_2$, pH = 2 ± 0.1, 100 °C | 15.78 | |
| | | | Sulfide fraction | HCl + $HNO_3$ (3:1 $v/v$), 120 °C | 56.30 | |
| | | | Residual | HF + HCl + $HNO_3$ | 5.89 | |

However, some problems were reported with earlier sequential extraction procedures, such as the non-specificity of the extracting agents and the reabsorption of metals before isolation for analysis [126–128]. It is also difficult to compare data obtained from different laboratories around the world that use different protocols. As a result, the BCR (now, the Standards, Measurement, and Testing Programme) established a BCR three-stage extraction method based on the Tessier method [129,130]. This method is operationally defined based on the extraction mechanism of the released metal rather than by discrete geochemical phases. At the same time, to strengthen the quality control of the analysis, the standard BCR601 for sediments was also developed. Moreover, the comparison results between 20 laboratories in eight EU countries also improved the accuracy and repeatability of the method. To standardize the sequential extraction scheme, the original BCR program has been modified [131] to divide the chemical species into acid-extractable, reducible, oxidizable, and residual fractions [132–134].

Based on the widely used Tessier or BCR methods, some other SE methods are also available with various combinations of leaching steps and sequences [24,53,125,135,136]. These methods were later modified and applied to soil or coal studies [136]. However, few of them were designed for the chemical speciation of V. Moreover, the few improvements that were made to the SE method were aimed at shortening the extraction time rather than improving metal recovery [35]. Unlike most other metal(loids) that generally exist either as anions or as cations in the soil, V geochemistry is very different. V may be present as both anions ($VO^{2+}$, $VO_2^+$) and cations ($HVO_4^{2-}$, $H_2VO_4^-$) [23]. Xu et al. established a new eight-step SE scheme that efficiently refined the V fraction bound to Mn, Fe, and Al (hydr)oxides and largely increased total extraction efficiency [24]. This was also the first study to enable the identification of visible amounts of geogenic V combined in the lattices of soil minerals.

SE can be used to provide an indication of the quantities of metals in various speciation, which is valuable for providing information on the mobility of V and other metals [15,28,29,137]. The existing research is mainly focused on assessing the potential risks of V in contaminated soils, such as identifying the amounts of anthropogenic V in contrast with V from natural origins [53,123,125], comparing the mobility of V in industrial or agricultural areas [27,28,35,51,73,91], and discussing the relationship between bioavailable V in soils and V contents in plants [23,25,36]. Recently, it has been found that SE methods using LMWOAs could reduce the mobilizable and bioavailable V in soils to achieve the effect of in-situ soil remediation [61,138].

### 3.3. Other Chemical Extraction Methods

Chemical methods are the most widely used methods for evaluating the bioavailability of metal pollutants and they are generally divided into two types, based on chemical extraction or mechanistic modeling [139]. In addition to common traditional single-step extractions and sequential extraction procedures, there are other effective chemical methods used to determine the bioavailability of elements in the environment in many cases [140,141].

Diffuse gradients in thin films (DGTs) are powerful in situ passive sampling techniques for performing analysis of metal species or speciation in soils, sediments, and waters [142,143]. DGT can be regarded as a passive sampler, mainly composed of a diffusion layer and a binding layer, which can extract targets from the environment [144]. Based on different binding gels, DGT can selectively accumulate different metal cations or oxyanions [143]. Four different DGT devices (Ferrihydrite, carbon, Chelex, and Purolite®) were used by Lucas et al. (2015) to determine the changes in the concentration of dissolved V and nine other elements (As, Au, Co, Cr, Cu, Co, Cr, U, Mo, As, Au, Zn, and Mn) in the water samples of an estuary [30]. Due to the formation of colloids or complexes bound to dissolve organic carbon (DOC), the DGT concentrations of V in the downstream site were lower than the total dissolved concentrations at the upstream site. In addition, changes in flow rates during different seasons can also affect the concentration of DOC, which in turn affects the DGT-dissolved V [31]. Metsorb-DGT and Ferrihydrite-DGT can be used to determine labile vanadates over a wide pH range and to accurately measure V(V) concentrations in seawater and freshwater [145,146]. An inverse trend was consistently observed between DGT-labile and dissolved V concentrations. The time of deployment, the concentrations of DOC, and the ionic strengths of the systems have subtle effects on either concentration [147].

Electrochemical techniques, such as scanned stripping chronopotentiometry (SSCP), absence of gradients and Nernstian equilibrium stripping (AGNES), anodic stripping voltammetry (ASV), and competitive ligand exchange or equilibration-cathodic stripping voltammetry (CLE-CSV) [34], are other powerful tools for metal speciation analysis. In the presence of high concentrations of added ligands, electrochemical techniques are very helpful in gaining insights into the cycling and potential bioavailability of various metals, such as Mn, Fe, V, Ti, and Cr, due to their low detection limits [148–152].

### 3.4. Physical Methods

Various anthropogenic and natural sources provide a gateway to release and introduce trace levels of V into the environment. Due to the carcinogenic and toxic effects of V, which might affect humans, plants and aquatic life, more efforts are need to determine low levels of V in natural waters and soil samples by using simple and easy methods [153]. V is usually present in trace quantities in different samples below the detection limit of most available instrumental systems. Thus, easy and simple sample preconcentration and separation technologies are necessary before screening out trace levels of V in different environmental samples [154–157]. Some physical separation techniques have consistently been reported to determine trace levels of V in different environmental samples.

Liquid-liquid microextraction (LLME) is one of the most appropriate extraction tools for separating toxic metals from complex samples [158]. Pekiner (2014) invented an in situ micropipette tip syringe system (μS-SHS) combined with ETAAS for separation, preconcentration and determination of V in food and water samples [155]. At pH = 6 and in the presence of interfering ions, high selectivity was shown for V in a surface aqueous solution with a vanadium concentration below $4.0 \ \mu g \cdot L^{-1}$ [155]. Alkali and alkaline earth elements do not form stable complexes with complexing agents, and they have almost no effect on the selectivity of the method at high concentrations [155]. As a portable method, it is suitable to separate trace amounts of different organic and inorganic toxicants from different environmental samples [159].

### 3.5. Spectroscopic Methods

X-ray absorption spectroscopy (XAS) is a powerful tool for determining the speciation of V present in sediments or soils [10]. In addition, X-ray absorption near edge structure (XANES) analyses can investigate the average valence state of numerous redox-active elements in solid samples and the species of V present in the soil in some cases [160,161]. Changes in the valence state of V from +3 to +5 result in corresponding shifts in pre-edge features and absorption edge positions. By fitting linear combinations of unknown species to a known reference, Larsson et al. (2017 a, b) calculated the average valence state of V in different mineral soils [162,163], and Burke et al. (2012) determined that V(V) dominated in red mud samples contaminated with As and Cr [164]. Through research on the V K-edge XANES spectra of highly weathered tropical soil samples and a range of reference compounds differing in V coordination chemistry or oxidation states, Wisawapipat and Kretzschmar (2017) also revealed that the majority of the V(IV)/V(V) is octahedral or tetrahedral. In limited cases, extended X-ray absorption fine structure (EXAFS) analysis has been used to determine the average molecular coordination environment of V in soil components [160]. In addition, extended X-ray absorption fine structure (EXAFS) measurements may be used to test the average molecular coordination environment of V in soil components [165–167]. By combining EXAFS analysis with ab initio molecular dynamics calculations, seminal studies were conducted on the binding of V(III, IV, and V) to gibbsite, goethite, ferrihydrite, and Fe(III)−natural organic matter complexes [160,162,168–170]. The use of V K-edge EXAFS spectroscopy is easily impeded by the Ba L2 edge, which strongly interferes with the V K-edge EXAFS region for sediments, slags, and soils [171,172]. Compared with other extraction methods, spectroscopic methods have a relatively high cost, and different methods are selected according to the actual situation [171]. Hence, EXAFS spectroscopy may have considerable limits as a tool for determining the average molecular coordination environment of V in many environmental materials, at least when using a conventional X-ray absorption setup.

### 3.6. Comparison of Different Extraction Methods of Vanadium

Conventional single extractant methods usually only have a good effect on one or two kinds of heavy metals, and it takes more time and cost to analyze the bioavailability of multiple heavy metals. At present, the development of analyses for multiple elemental states is growing; several single extraction *n* procedures can be applied to conduct extraction tests to optimize the extraction conditions of the extracting agents for multiple elemental states. In contrast to sequential extraction methods, single extraction methods suffer from the difficulty of discovering a single reagent that can quantitatively dissolve the residual species of metals without attacking the detrital species. Sequential extraction methods are too cumbersome for studying the bioavailability of heavy metals, and the definition of morphological classification and bioavailability is slightly different. Some scholars often use fast and effective single extraction methods to analyze the effectiveness of V. Yang (2015) showed that the correlation between sodium nitrate and alfalfa in the speciation extraction of V was better than that of the BVR method and EDTA method [25]. DGT technology is hardly affected by the basic properties of soil and can predict the availability of V in soil well, but DGT technology cannot fully simulate the plant growth environment [173]. The Tessier method of extractants lacks selectivity, and there are resorption and redistribution phenomena during the extraction process, a lack of validation, and low data accuracy [174]. On this basis, BCR strengthened the control of analytical quality and formulated reference material (Certified Reference Material, CRM 601) [132]. The three-step BCR method fully utilizes the selective extractant from weak to strong, minimizing phase channeling [175]. By analyzing the phase state of sludge from WWTPs, the effects of the BCR method and Tessier method were compared, and the results showed that BCR was more effective than Tessier in extracting oxidizable states [176].

## 4. V Bioavailability in Soils Based on Morphological Extraction

### 4.1. Plant-Available V in Soils

V acts as a growth-promoting factor, which could improve nitrogen assimilation and utilization, chlorophyll biosynthesis, and seed germination. Under V stress, Mo and B concentrations decrease in roots and increase in upper leaves [177–179]. Both of these metal concentrations closely correlate with nitrate reduction, which might be responsible for the increased nitrogen levels in the leaves of V-treated plants [180]. Furthermore, V also enhances the uptake of Fe and Mg, which are essential elements for chlorophyll biosynthesis [14,181]. Therefore, V can promote root length, plant height, and biomass production at low contents. However, high V levels may inhibit key enzymes that mediate ion transport, protein synthesis, energy production, and other important physiological processes and cause root and shoot abnormalities, growth retardation, and even mortality in plants [14,182,183]. V causes reductions in carotenoid content, potassium (K) uptake and transport, and other growth factors, which could reduce photosynthetic activity and transpiration [181,184,185].

V has a complex species in soil and has various oxidation states (+2, +3, +4, +5), but it exists mainly in the species of V(IV) and V(V) in soil [5,85]. Accumulation is the major response strategy of most plants to the soil toxicity of V. The available species of V that can be absorbed and utilized by plants in the soil include those that are acid-extractable, reducible, and oxidizable. Most of the V absorbed by plants is accumulated in the roots, and a small part is transferred from roots to the trunk and leaves. The process of V absorption by roots and transport in the plant is shown in Figure 1 [14]. V in the movable part of the soil is mainly pentavalent [85]. After entering the plant, the pentavalent V is gradually transformed into tetravalent V by the reduction of plant cells [186]. V is mainly concentrated in the roots, where V contents are approximately 2 to 1000 times higher than its levels in aerial parts of plants [23]. The amount of V in soil is the most important factor influencing V accumulation in roots below the threshold V levels of apparent toxicity to plants [7]. The half-maximal effective concentration (EC50) is a toxicological index that can be used to counter the relationship between heavy metals and acute toxicity. $EC_{50}$ of V for plants grown under hydroponic conditions varied from 1 to 50 mg/L, while it varied from 18 to 510 mg/kg in soils. In many cases, a variety of plants exhibited strong tolerance to V, such as some legumes (crops of chickpea, soybean, green bean, alfalfa, etc.) and vegetables (Chinese green mustard, tomato, rice, lettuce, etc.) [7,23,25,38,184,187–190]. The greatest known V accumulation was detected in the tissues of Chinese green mustard, bunny cactus, and chickpea at over 8000 mg/kg [25,38,187,190]. As a result, these plants are suitable for V-contaminated soil remediation on a large scale. The bioavailable V in the soil enters the plant through the root system. However, through the food chain, this effect can be transmitted to people, which has an impact on human health. Soil V content associated with higher gastric and colorectal cancer mortality rates in humans (IPCS, 1998). In the presence of $H_2O_2$ at the site of inflammation, V activates mast cells at the late phase to amplify allergic responses [191]. V(IV) can oxidize a large number of biochemical substances, generate free radicals, and cause DNA damage [192].

Under conditions of high environmental V concentrations, Plants and root microorganisms secrete some organic substances to change the physical and chemical properties of the environment (pH, Eh etc.) to resist V [193,194]. In addition, Most of the V absorbed by plants from the environment forms coordination bonds with polysaccharides and hydroxyl groups in plant cell walls and cannot enter plants [180]. some plants have low absorption characteristics for V. Rape, watermelon, box-thorn, and Chinese cabbage accumulate only small amounts of V (0.56 mg/kg, 1.3 mg/kg, 2.73 mg/kg, 3.00 mg/kg, respectively) in their roots [183,193,195,196], revealing high-efficiency exclusion and elimination properties against high V contents. From the perspective of efficient land use and food safety, some plants have some resistance to V-contaminated soil, but whether there is any negative impact on health needs to be studied in more detail.

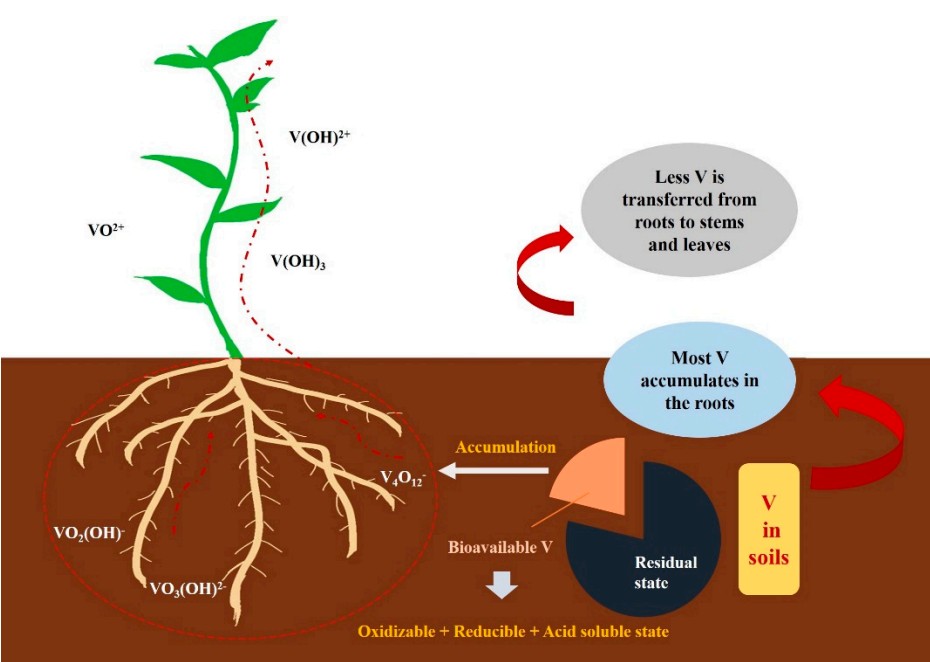

**Figure 1.** The absorption and transformation of V in plants.

*4.2. Bioavailability Evaluation Index or Method*

There is general consensus on the bioavailability of heavy metals that can assist the risk assessment of these elements in the environment. However, professionals from different disciplines can have different understandings of and definitions for bioavailability. The International Union of Pure and Applied Chemistry (IUPAC) defines bioavailability as the bioavailable part of a chemical "able to be absorbed by living organisms", which is also a function of biological properties and chemical speciation [197]. Bioavailability should be quantifiable in risk assessment, but there are no unified analysis methods to quantify it. In general, chemical methods and bioassays can be used to assess potential bioavailability in extracellular or intracellular matrices, respectively [198–201]. Among these methods, morphological extraction mainly uses singly coupled or sequential extraction reagents to simulate different redox environments to predict risks related to contaminant behavior under specific environmental conditions. Morphological leaching tests are relatively simple and reproducible methods that can be performed according to standardized protocols, so they are the most widely used methods for evaluating the bioavailability of heavy metals (e.g., US EPA, 1992; DIN, 1998; US NRC, 2002) [202–204].

Available V in the soil refers to the content of V that can be quickly absorbed and assimilated by plants. Pot experiments have showed that the growth of plants substantially reduces the concentrations of V(V) in the rhizosphere soil, but no such relationships were found for V(IV), suggesting that V(V) is actively related to the soil–root interaction of V [19,23,182]. The bioavailability of V(V) is affected by soil pH value, total organic carbon (TOC), plant species, and V concentration in soil [2]. V forms complex with carboxylic acid and glycooh groups, and exists in the species of $VO^{2+}$ under strong acidic conditions [186]. V at neutral pH exists mainly in the species of tetravalent V cation ($VO^{2+}$) and pentavalent vanadate anion ($HVO_4^{2-}$ or $H_2VO_4^{-}$) [205,206]. There was a poor correlation between plant biomass and soil V(V) at low concentrations in the soil, while these parameters showed a negative correlation at higher concentrations, reflecting the toxic effect of V [182]. The presence of carbonate plays an important role in vanadium mobility [15]. Therefore, these results indicate that V(V) concentrations might better reveal the toxicity of V in soils than total V or soil V(IV), which can be used as an indicator of V bioavailability [94]. Moreover, the self-protective function of plants might prevent the translocation of V from the root to the aboveground parts [193,207]. Thus, the roots always absorb more V than the

other parts of the plant (e.g., stem, leaf, and seed) [23,182]. Furthermore, the contents of total V in roots are consistently proportional to the water-soluble or extractable V concentrations in the rhizosphere soil [208]. For this reason, these V fractions that are effective for plants can be used to indicate the bioavailability of V in the soil-plant system, to a certain extent.

From the perspective of soil chemistry, the bioavailability of V not only includes water-soluble, acid-soluble, chelated, and adsorbed states but also includes forms that can be released into plants in a short period, such as some easily decomposed organic states and weathered mineral states. Sequential extraction methods are usually used to quantify V fractions that can be mobilized in acidified, reduced, or oxidized environments. By the standardized Tessier five-step sequential extraction or modified BCR method, bioavailable V could be defined as the sum of the first few steps excluding the residual phases [25,36,50]. Many research results have shown that V sorbed by plants has a direct correlation with V in soils, especially for the sum of the first three fractions detected by BCR methods [16,23,25,209].

Single extraction methods can be used directly to predict V bioavailability in soils because they have good correlation with plant uptake, and are usually used to evaluate different hazardous species. Weak acids (e.g., 0.1 M HOAc, pH = 3.5) are expected to minimally influence the extraction of V and to predominantly release only the weakly adsorbed V fractions, which better reveal the readily bioavailable species in soils [182]. HCl and EDTA extractions seem to reflect long-term influences because they may decrease the adsorption affinity of V by dissolving amorphous soil minerals, especially Fe and Al (hydr)oxides, which are strong ligands for V complexation [210,211]. $CaCl_2$ and $NaNO_3$, which are regarded as soil background electrolyte solutions, could be adopted as extractants for V bioavailability prediction. Low-molecular-weight organic acids (LMWOAs) (e.g., citric, malic, acetic, lactic, and formic acids, etc.), which are produced in the rhizosphere environment, are secretions of fungi and bacteria [100,101]. They play an important role in transporting metals to roots and improving the uptake of metals by plants, so they are good indicators of the bioavailability of V operating at the soil-root interface [211–213]. Nevertheless, no single specific extractant can be used as a standard in a universal method for predicting bioavailability [214]. The extractability of V in soils varies with extraction reagents, which can be explained by different extraction mechanisms.

## 5. Conclusions and Perspectives

In summary, we briefly reviewed the distribution of global V-bearing minerals and the main sources of V pollution in the environment. Then, based on physical and chemical methods, different extraction technologies and their extraction efficiencies for V in the environment were discussed. The general conclusions of these comparisons follow: (i) Single extraction methods can be widely used to evaluate the short-term or medium-term hazards of V in soils and sediments because the extracted chemical species correlate well with V content in plants. (ii) SE, which is mainly used to assess the potential risk of V in contaminated soils, can be used to estimate the amounts of V in various "reservoirs," but challenges remain with regard to the non-specificity of extracting agents, incomplete extraction between different phases, and reabsorption of metals before isolation for analysis. (iii) Electrochemical techniques, such as SSCP and ASV, are suitable for studying the circulation of heavy metals such as V and their potential bioavailability under high concentrations of ligand addition. (iv) Physical methods can be used for the separation, pre-concentration, and determination of trace amounts of V in food and water samples; these techniques have high selectivity for V in the presence of interfering ions. (v) XAS is also a powerful tool to determine the species of V in solids because XANES analysis can investigate the average valence of multiple redox active elements and ligand species of V(IV)/V(V), and EXAFS measurement results can be used to test the average molecular coordination environments of V in soil components.

Studies have shown that the chemical species of V and its bioavailability cannot be attributed to a single value that can be measured by a single chemical or even biological method. As with any process in nature, it is dynamic and changes with time and envi-

ronmental conditions. Many efforts have been made recently to correlate bioavailability with chemical extractions results. To reliably assess and predict the long-term behavior of V in the environment, a complete set of detailed biological and chemical tests coupled with geochemical modeling and advanced spectroscopy techniques may be required. In addition, it is impractical to conduct extensive investigations on the chemical species of V in the field. A small set of tests can be collected for experimental verification to more accurately and conveniently predict changes in V bioavailability.

**Author Contributions:** Conceptualization, Methodology, Writing—Original Draft, J.Y.; Writing—Original Draft, Software. Y.W.; Conceptualization, X.G.; Resources, Investigation, Data Curation, C.J.; Software, Supervision, R.Z.; Project administration, Supervision J.W.; Formal analysis, Supervision, L.S.; Writing—Review & Editing, Funding acquisition, Y.T. All authors have read and agreed to the published version of the manuscript.

**Funding:** This work is funded by [the National Natural Science Foundation of China] grant number (41403085), [the National Natural Science Foundation of China] grant number (41877355) and the National Key R&D Program of China] grant number (2018YFC1800901).

**Data Availability Statement:** The data presented in this study are available in insert article here.

**Acknowledgments:** We appreciate two anonymous reviewers for critical comments that further refine the quality of the article.

**Conflicts of Interest:** The authors declare no conflict of interest.

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
