# Peer review of "Vanadium: A Review of Different Extraction Methods to Evaluate Bioavailability and Speciation"

_minerals, doi:10.3390/min12050642_

Round 1

Reviewer 1 Report

This is a V review paper mainly focusing on extraction methods to evaluate V speciation and bioavailability. The title should be changed to represent that "Vanadium: A review of different extraction methods to evaluate bioavailability and speciation".

It is a timely review since V is re-emerging as a toxic metal. In the beginning, it was not clear author's focus was on soils. Since plant bioavailability is the focus of the abstract, the authors should clearly state that they are evaluating soil extraction methods. In some places, they are talking about extractions of V from fly ash and other materials, but that is not relevant. Better to focus on soil and sediments only. Secondly, in many places, they are trying to generalize findings to toxic metals or heavy metals. That is again not necessary, therefore, narrow down only to Vanadium.

V speciation should be discussed under one subtitle elaborately. That is the main part of your title

Extraction methods, bioavailability, and speciation are equally important in this review. Authors gave priority and focused mainly on extraction methods other two sections touched barely.

In the plant-available V section emphasis was given to phytotoxicity of V and phytoremediation possibilities. Is that the objective of this paper? Minimum attention was given to food-chain transfers to animals and humans. This should be included in the review section of bioavailability.

I made more detailed comments (74) in the text, please attend carefully.

Author Response

Thank you for the beneficial comments and suggestions on each section of this manuscript. The corresponding changes have been made in the revised manuscript according to your comments point-by-point:

1、This is a V review paper mainly focusing on extraction methods to evaluate V speciation and bioavailability. The title should be changed to represent that "Vanadium: A review of different extraction methods to evaluate bioavailability and speciation".

Based on the suggestions of reviewers, we carefully revised the title of the article to make it more in line with the content.

2、It is a timely review since V is re-emerging as a toxic metal. In the beginning, it was not clear author's focus was on soils. Since plant bioavailability is the focus of the abstract, the authors should clearly state that they are evaluating soil extraction methods. In some places, they are talking about extractions of V from fly ash and other materials, but that is not relevant. Better to focus on soil and sediments only. Secondly, in many places, they are trying to generalize findings to toxic metals or heavy metals. That is again not necessary, therefore, narrow down only to Vanadium.

Thank you for your valuable suggestion. I have carefully revised the content of the article to make it clear that the research focus of the article is to evaluate the extraction method of vanadium bioavailability in soil. Content not relevant to the title of this article, such as the extraction of vanadium from fly ash and other materials, has been removed. The current article focuses only on soils and sediments. Regarding generalization to other heavy metals or toxic metals, we have made changes to only keep the content related to vanadium.

3、V speciation should be discussed under one subtitle elaborately. That is the main part of your title

Regarding the content of V speciation, we have added a new subtitle elaborately “V speciation in soil” that describes the content of V speciation in detail.

4、Extraction methods, bioavailability, and speciation are equally important in this review. Authors gave priority and focused mainly on extraction methods other two sections touched barely.

We appreciate for your valuable comment. we have added a new subtopic V speciation to the article to give a detailed introduction to the speciation of vanadium. At the same time, the content on bioavailability is supplemented.

5、In the plant-available V section emphasis was given to phytotoxicity of V and phytoremediation possibilities. Is that the objective of this paper? Minimum attention was given to food-chain transfers to animals and humans. This should be included in the review section of bioavailability.

The content of vanadium toxicity and phytoremediation possibilities has been omitted. We have carefully reviewed the literature on the transfer of vanadium to animals and humans through the food chain and have described it in detail in the plant-available V section.

6、I made more detailed comments (74) in the text, please attend carefully.

Thank you for your valuable comment. 

Reviewer 2 Report

This review manuscript is well organized and written in English. In addition, the topic is interested in the field of analysis and assessment of metal speciation. Therefore, it can be published after minor revision after followed comments.

  1. In this manuscript, most of the data have focused on investigation in China. So, the author needs to add proper data from other countries.
  2. In the manuscript, a comparison of the various extraction methods of V is requested to understand the intractability and usage of single extraction methods and sequential extraction methods.

Author Response

Thanks for your valuable comments and suggestions. We have accepted all of your comments. As to points by you, we would like to reply as follows:

1、In this manuscript, most of the data have focused on investigation in China. So, the author needs to add proper data from other countries.

We appreciate for your valuable comment. We carefully supplemented the average concentration range data in the soil of many countries with large vanadium reserves, such as Russia, and presented the relevant content in the table 1.

2、In the manuscript, a comparison of the various extraction methods of V is requested to understand the intractability and usage of single extraction methods and sequential extraction methods.

We listened carefully to your suggestion, and added a subtitle of “a comparison of the various extraction methods of V” to the article to compare the various extraction methods of V.

Round 2

Reviewer 1 Report

I am glad to see the improvements in the paper. There are some running sentences to be corrected. I made a few more corrections to the manuscript. Good luck!

Author Response

Answer to Reviewer:

Thank you for the beneficial comments and suggestions on each section of this manuscript. The revised manuscript has been revised point by point according to your comments, and the revised places are marked with red fonts. The points that need to be explained are listed below.

1、I think better to stay with "species" and please check for the consistence in whole paper - form is misleading, isnt it?

Thank you for your comments, we checked the whole article and replaced the inappropriate word “form” with “species”. At the same time, we have revised other inaccurate words in the article.

2、Are there any evidences for V in animal supplements? Validity of the paper increase if you include the that reference, not on toxi metals in general.

We appreciate for your valuable comment. Through the search of relevant references, there is no clear evidence that V is used as an animal supplements. we have deleted some of the content and only kept the content related to vanadium.

3、better to use same term as sequential

Based on the suggestions of reviewers, we checked the relevant words in the full text and made changes to ensure the consistency of words such as sequential extraction methods.
